# Adolescent HIV prevent and care framework: A global scoping review protocol- BSGH 006

**Gamji Rabiu Abu-Ba'are**[1,2,3,4,5,6]◉, **Osman Wumpini Shamrock**◉[1,2,3]◉*,
**Darcey Rodriguez**[7], **George Rudolph Kofi Agbemedu**[2], **LaRon E. Nelson**[2,3,4,5]

**1** School of Nursing, University of Rochester, Rochester, New York, United States of America, **2** Behavioral, Sexual, and Global Health Lab, University of Rochester, Rochester, New York, United States of America, **3** Behavioral, Sexual, and Global Health Lab, Accra, Ghana, **4** School of Nursing, Yale University, New Haven, Connecticut, United States of America, **5** Center for Interdisciplinary Research on AIDS, Yale University School of Public Health, New Haven, Connecticut, United States of America, **6** Department of Public Health Sciences, University of Rochester Medical Center, Rochester, New York, United States of America, **7** Edward G. Miner Library, University of Rochester Medical Center, Rochester, New York, United States of America

◉ These authors contributed equally to this work.
* osmanwumpini_shamrock@urmc.rochester.edu

**Data Availability Statement:** Deidentified research data will be made publicly available when the study is completed and published.

## Abstract

Among adolescents, HIV/AIDs remains a significant cause of death globally [1–4]. Given the unique stages in human development, adolescents have been shown to fall within a sexually active phase. Combined with other social and structural factors in their immediate environments, HIV prevention and care among adolescents can be filled with challenges for intervention. This paper outlines this protocol to systematically review peer-reviewed literature in prevention and care among adolescents 10–19 years. The Preferred Reporting Items for Systematic Reviews and Meta-Analyses extension for Scoping Reviews (PRISMA-ScR) will be used to report this scoping review. The review will involve screening and extracting data using covidence as the primary tool. The review will encompass quantitative, qualitative, and mixed methods studies, utilizing a search strategy from electronic databases such as PubMed (NCBI), Web of Science Core Collection (Clarivate), Embase (Elsevier), and Scopus (Elsevier). Additionally, a search will be conducted for grey literature using Global Index Medicus (WHO), MedNar (Deep Web Technologies), and Central Register of Controlled Trials (Cochrane). Duplicate removal and selection of articles that meet the inclusion criteria for the study will be performed using Covidence. Once the screening process is complete, data will be extracted from the full-text screened articles in Covidence. We will pilot the extracted data in Covidence to ensure that all relevant information has been captured, making necessary changes if required. Data extraction will be carried out by at least two authors, with any conflicts resolved by the same authors. If a conflict cannot be resolved between the two, a third author will make a final determination. We aim to analyze data thematically by employing a grounded theory approach to generate codes pertinent to the research question. The team will review and discuss codes to create a cohesive set of codes that will be instrumental in identifying knowledge gaps and constructing themes that summarize the data. The proposed systematic review will be among the pioneering efforts to rigorously assess global data on HIV prevention and care, with a specific focus on

**Funding:** The authors received no specific funding for this work.

**Competing interests:** The authors have declared that no competing interests exist.

adolescents 10–19 years. It will consider the diverse socio-economic factors and experiences shaping these adolescents' lives in HIV prevention and care. We expect this review to yield critical insights into the present landscape of HIV prevention and care for individuals aged 10–19. These findings will also play a pivotal role in shaping the development of a global framework that researchers and stakeholders can readily adopt and implement across socio-economic contexts. This framework will aim to address the unique needs of all adolescents concerning HIV prevention and care.

## Introduction

Globally, HIV remains one of the leading causes of death worldwide [5]. Controlling the spread of the virus has been a major challenge, posing significant problems in advancing preventative measures and providing continuous care to affected individuals [6–10]. The global prevalence of HIV has reduced in this decade, but significant regional variations in trends and modes of transmission exist [11, 12]. The epidemiology of HIV among various key population groups, such as adolescents and socially marginalized groups, based on varying community perceptions about sex, has challenged the global efforts to control the spread of the virus [13–17]. Adolescents aged 10–19 years account for approximately 16% of the world's population. In 2021, 1.7 million adolescents were living with HIV, comprising 5% of all people living with HIV globally, and accounted for 11% of all new infections among emerging adults [18, 19].

Adolescent HIV/AIDS is a distinct epidemic that needs to be handled and managed differently from adult HIV [20, 21]. Adolescent sexual and reproductive health remains a significant public health concern globally, especially in communities where child marriage and adolescent childbearing are high, with low exposure to modern contraceptives [22]. Several factors contribute to the high prevalence of HIV among adolescents, including vulnerability during their transitional stage in life and biological changes that can affect their social relationships, potentially leading to risky behaviors that increase the risk of HIV infections [19, 23–25]. The severity of addressing HIV prevention and care in adolescence is crucial, as many adolescents with HIV are unaware of their status and fail to consistently use condoms in sexual encounters or face challenges when accessing preventive options due to alcohol-drug abuse, sex-HIV-age stigma and discrimination [26–27].

Adherence to HIV medication is a concern among adolescents diagnosed with HIV [28–32]. Accepting a positive HIV test result is challenging for adolescents due to the need for life-long treatment and worries about their future goals regarding health, education, career and marriage, among other personal goals, producing a micro-level barrier to adolescents' readiness for HIV care [21, 33]. HIV stigma and sex stigma in some communities deter adolescents from accessing HIV care [33, 34]. Additionally, the side effects of HIV medications can affect adherence as they can affect adolescents' ability and willingness to consistently take their prescribed Antiretrovirals [35–37]. Determining appropriate antiretroviral (ART) dosages for sustained suppression in adolescents is challenging adolescent may experience growth spurt, particularly in under resourced health facilities [38]. Failure to take the correct dosage of antiretroviral can intensify side effects and result in poor adherence [39–41].

Although several preventative measures have been proposed to curb the incidence of HIV among adolescents, it has been recognized that no singular technique is enough to address the epidemic adequately, hence, an integrated approach that combines biomedical preventive techniques with behavioral and structural interventions is recommended as the ideal means of

preventing HIV among adolescents [6, 42]. The World Health Organization has also proposed using psychosocial techniques to support preventive measures for HIV among adolescents [43]. Considering that contextual factors significantly influence the effectiveness of interventions in HIV prevention and care among adolescents, it is important to pay attention to regional dynamics when proposing interventions for adolescents. Adolescent girls in 2021 accounted for 75% of all new HIV infections among adolescents. This percentage was even higher with adolescent girls in the 35 HIV-priority countries in UNICEF's Strategic Plan, accounting for 80% of all new HIV infections among adolescents. In sub-Saharan Africa in the same year, about six times as many adolescent girls were newly infected with HIV than adolescent boys. Beyond the sub-Saharan Africa region, the highest numbers of HIV-positive adolescents are in Asia and Latin America. In East Asia and the Pacific, more boys are newly infected with HIV each year than girls in adolescence. This finding mirrors the various distinct risk behaviors in the respective regions, which suggests that interventions must be personalized to the specific nature and dynamic of the epidemic.

Moreover, there's a lack of a universal framework that adequately encompasses the multifaceted dynamics of HIV among adolescents, particularly those between the ages of 10–19 years. It's important to address the gaps in adolescent HIV prevention and care research. Failing to consider social differences, intersectionality, and political climates in different geographical areas could hinder the development of practical and universally applicable policy recommendations that cater to the needs of this population. Our review proposes to:

1. Develop a comprehensive assessment of data by conducting a thorough evaluation of existing research on HIV prevention and care among adolescents aged 10–19, considering diverse social factors such as geography, race, sex, age, gender, education, religion, sexual orientation, occupation, nationality, marital status, socio-economic status, disability, and immigration status.

2. Use data to develop a global framework based on theory, research and practice to capture the diverse dynamics inherent in adolescent HIV prevention and care. This framework will be adaptable to various economic contexts and provide a standardized measure to support intervention in adolescent HIV prevention and care.

## Method and analysis

We will utilize the guidelines provided by the Preferred Reporting Items for Systematic Reviews and Meta-Analyses Extension for Scoping Reviews (PRISMA-ScR) by Tricco et al. (2018) for reporting this scoping review. As outlined by Tricco et al. (2018), the first step in conducting a scoping review is to develop a protocol to clarify the purpose and methodology of the review. By documenting and sharing this protocol, we aim to ensure transparency in our process and prevent duplication of efforts in the field of adolescent HIV prevention and care [45, 46].

## Ethics and dissemination

Our scoping review findings will be published in a peer-reviewed journal, making them accessible to researchers, practitioners, and the public. We will also share these findings at relevant conferences and events on adolescent HIV prevention and care. Additionally, we aim to use the insights gained from this review to inform future research studies, addressing gaps in the scholarly literature on HIV among adolescents, particularly in prevention and care.

## Patient and public involvement

This study does not involve human participants or data collection, so it does not require ethical approval. The research findings will be disseminated through various channels, including conferences, webinars, and peer-reviewed manuscripts. The insights gained from this study have the potential to inform and influence the policies and practices of government health agencies and healthcare facilities, fostering improvements in the field.

## Criteria for study inclusion

Only articles addressing HIV prevention or care among adolescents aged 10–19 will be considered. Such articles must present empirical data on the topic and be published in English. We will limit the search inclusion to English for two reasons. The first is due to the unavailability of a multilingual reviewer in the research team and the cost of translation services. The second is connected to our researchers' expertise and proficiencies in English to ensure an accurate and thorough review of selected studies, enhancing the reliability of data extraction and minimizing the possibility of misinterpretation. Only studies published after 2013 will be included to provide up-to-date information.

## Criteria for study exclusion

The review will exclude all individuals aged below 10 and above 19 years. Studies that do not capture empirical data on the topic and are not published in English will be excluded. Studies published before 2013 will be excluded. This review will not include certain types of publications such as review papers (including scoping and systematic reviews), book chapters, reports, opinions, commentaries, conference abstracts, and articles published in languages other than English.

## Types of studies

Our review will include quantitative, qualitative, and mixed methods studies. We will encompass both experimental and observational studies without excluding any based on methodological approaches. However, we will exclude articles that synthesize existing literature, such as reviews, in our analysis.

## Search strategy

**Identifying sources.** A medical librarian (DR) from the University of Rochester Medical Center will create the search strategy with input from other authors to help find any extra sources, such as grey literature, similar to what was done in other protocols [45, 46].

**Electronic database searching.** DR will conduct a literature search using PubMed (NCBI), Web of Science Core Collection (Clarivate), Embase (Elsevier), and Scopus (Elsevier). The search strategy will use a combination of index terms when available in each database and keywords including HIV, adolescent, care continuum, and anti-retroviral agents. See attached file for an example. The search will limit results to those published in the English language between 2013 to present.

**Grey literature searching.** Grey literature will also be searched for using Global Index Medicus (WHO), MedNar (Deep Web Technologies), and Central Registry of Controlled Trials (Cochrane) to capture other relevant data that may not be included in other databases.

**Data screening.** After the search process is complete, all articles will be exported to Covidence, and duplicate articles will be removed. Using the inclusion criteria set forth by the team, a two-step screening process will occur in Covidence. After de-duplication, the first

process will include two reviewers screening the title and abstract of all remaining articles. After the title and abstract screening process, a team of two will independently review the full text of articles that have met the inclusion criteria. When the two reviewers complete the screening process, they will meet to resolve any conflicts. If the two reviewers cannot resolve, a third reviewer will make a final determination.

## Data extraction

**Content.** Data extraction will begin after all articles have undergone title, abstract, and full-text screening. Using Covidence, the data extraction form will identify publication details, including title, author, and year of publication. The form will also extract data on the methods, including study design, aim, and date. Participate data will include a description of the population, methods of recruitment, and number of participants.

**Process.** To ensure that all the relevant data is captured as intended, the data extraction form in Covidence will be pilot-tested on a few studies. After pilot testing, modifications will be made, and the form will be used to collect data from eligible studies. The data extraction process will involve two authors working together to ensure accuracy. In the event of any conflicts, they will discuss and resolve the issues. If they are unable to reach an agreement, a third reviewer will be consulted to make a final determination. The review will adhere to the JBI guidelines when conducting the review.

**Analysis and reporting.** The findings in the study will adhere to the guidelines outlined in PRISMA-ScR [44]. We will present a narrative summary of the results while utilizing tables to organize the data. The outcomes will be categorized, taking into account the number of studies, their designs, and methodology. Additionally, key findings from each study will be condensed and presented in tables. Although we will focus on screening experimental studies, we will not conduct quantitative data analysis. Instead, we will employ descriptive statistics such as frequency and range to elucidate the results. Our data will be thematically analyzed [45, 46]. We will utilize grounded theory and establish a comprehensive list of codes relevant to the research question and outcomes. This code list will undergo a duplicate review by the research team involved in the data extraction process. A consensus will be reached on a unified set of codes through group discussion. These codes serve as the foundation for creating themes that capture the narrative synthesis of the extracted data and identify any existing knowledge gaps. The proposed framework will be guided by data derived from the systematic review and recommendations from researchers in the diverse research fields in HIV prevention and care among adolescents aged 10–19 years. We will use data derived to serve as a checklist and measure the extent to which the framework considers the social differences, intersectionality, and political climates of adolescents globally.

**Outcome.** The primary objective of this study is to gain a comprehensive understanding of the current state of HIV prevention and care research among adolescents aged 10–19 years. We will use this knowledge to create a global framework that can be easily used in various economic contexts and serve as a standardized intervention measure in adolescent HIV prevention and care. This framework proposes to enhance research, theory, practice, and policy efforts that consider the diverse experiences of adolescents in HIV prevention and care.

## Discussion

The prevention and care of HIV among adolescents pose significant global challenges due to various factors, including their transitional stage in life, vulnerability associated with their age groups, and the complex interplay of biological, structural, and social factors [23, 24, 47]. These factors individually and collectively may hinder efforts to reduce adolescent HIV-related

deaths and transmission. The prevalence of HIV among adolescents is particularly concerning, with areas such as the sub-Saharan region experiencing higher burden of HIV-related deaths among this population group, exacerbated by risky behaviors associated with their transitional stage and a lack of awareness of HIV status [4]. Our review will aim to analyze global data on HIV prevention and care, focusing on adolescents aged 10–19 years. We will take into account various socio-economic factors and life experiences that affect how these adolescents engage with HIV prevention and care. The insights gained from this analysis will be invaluable in tailoring HIV prevention and care specifically for this age group. Additionally, we plan to use these findings to create a framework that researchers and stakeholders can easily adopt in different socio-economic contexts. This framework will cater to the unique needs of all adolescents in HIV prevention and care, ensuring inclusivity and effectiveness in diverse settings.

## Strengths and limitations of this study

1. A notable strength of this study will be its rigorous approach to literature search conducted by a skilled librarian. The search strategy will be comprehensive, augmented by including grey literature sources. This meticulous process will thoroughly cover the existing literature, bolstering the study's credibility and reliability.

2. Furthermore, this study will offer significant advantage by providing cutting-edge insights and valuable guidance for future research and interventions focused on enhancing awareness among adolescents in HIV prevention and care globally. By keeping abreast of the latest developments, this research will actively advance effective strategies and interventions in this critical area of HIV research.

3. The restriction to articles published exclusively in the English language may be one of the notable limitations of this study. While this ensures consistency and enables a comprehensive analysis within that language, it may inadvertently exclude valuable contributions from non-English sources.

4. Another anticipated limitation of this study will be its specific focus on adolescents (10–19) as the target population. While this emphasis allows for in-depth exploration and tailored insights, it may overlook valuable perspectives and findings relevant to other age groups or people affected by HIV prevention and care.

## Supporting information

**S1 Checklist. PRISMA-P 2015 checklist.**
(DOCX)

## Author Contributions

**Conceptualization:** Gamji Rabiu Abu-Ba'are, Osman Wumpini Shamrock, George Rudolph Kofi Agbemedu, LaRon E. Nelson.

**Methodology:** Darcey Rodriguez.

**Supervision:** Gamji Rabiu Abu-Ba'are, Osman Wumpini Shamrock.

**Writing – original draft:** Gamji Rabiu Abu-Ba'are, Osman Wumpini Shamrock, Darcey Rodriguez, George Rudolph Kofi Agbemedu, LaRon E. Nelson.

**Writing – review & editing:** Gamji Rabiu Abu-Ba'are, Osman Wumpini Shamrock, Darcey Rodriguez, George Rudolph Kofi Agbemedu, LaRon E. Nelson.

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
