## [Decision Letter · Decision Letter 0]

25 Oct 2023

PONE-D-23-24131Adolescent HIV Prevent and Care Framework: A Global Scoping Review ProtocolPLOS ONE

Dear Dr. Shamrock,

Thank you for submitting your manuscript to PLOS ONE. After careful consideration, we feel that it has merit but does not fully meet PLOS ONE’s publication criteria as it currently stands. There are some methodological and structural issues that require attention. Also the protocol is written in past tense making one to wonder if this work has already been completed. We therefore, invite you to submit a revised version of the manuscript that addresses the points raised during the review process. Please submit your revised manuscript by Dec 09 2023 11:59PM. If you will need more time than this to complete your revisions, please reply to this message or contact the journal office at plosone@plos.org. Please include the following items when submitting your revised manuscript:A rebuttal letter that responds to each point raised by the academic editor and reviewer(s). You should upload this letter as a separate file labeled 'Response to Reviewers'.A marked-up copy of your manuscript that highlights changes made to the original version. You should upload this as a separate file labeled 'Revised Manuscript with Track Changes'.An unmarked version of your revised paper without tracked changes. You should upload this as a separate file labeled 'Manuscript'.

We look forward to receiving your revised manuscript.

Kind regards,

Edward Nicol, PhD

Academic Editor

PLOS ONE

Journal Requirements:

Reviewers' comments:

Reviewer's Responses to Questions

**Comments to the Author**

1. Does the manuscript provide a valid rationale for the proposed study, with clearly identified and justified research questions?

Reviewer #1: Yes

Reviewer #2: No

2. Is the protocol technically sound and planned in a manner that will lead to a meaningful outcome and allow testing the stated hypotheses?

Reviewer #1: Yes

Reviewer #2: Partly

3. Is the methodology feasible and described in sufficient detail to allow the work to be replicable?

Reviewer #1: Yes

Reviewer #2: Yes

4. Have the authors described where all data underlying the findings will be made available when the study is complete?

Reviewer #1: No

Reviewer #2: Yes

5. Is the manuscript presented in an intelligible fashion and written in standard English?

Reviewer #1: Yes

Reviewer #2: Yes

6. Review Comments to the Author

You may also provide optional suggestions and comments to authors that they might find helpful in planning their study.

Reviewer #1: An important review. Suggest the following to improve the protocol.

(BSGH) code in the manuscript title needs to be removed.

Abstract:

Brief the article selection and initial screening. It appears that authors intend to perform initial screening just with Covidence (‘Duplicate removal and selection of articles that meet the inclusion criteria for the study will be performed using Covidence’). Covidence would be the platform to facilitate the process.

Not clear as to what is exactly expected by mentioning ‘The extraction would include information on abstracts ….’. The data extraction needs to be from the full article content.

The abstract ends abruptly. Please include briefings of data analysis and presentation, and discussion in the abstract.

Main manuscript:

Strengthen the justification, late in the introduction, by highlighting the actual knowledge gap and it’s importance so emphasizing why this review needs to be conducted.

Present the review objective(s) / questions in a precise manner, as it appears to be much broader focusing on different aspects. Better to present the precise objectives separately.

Indicate the adherence to JBI guidelines in conducting the review, apart from reporting according to PRISMA-ScR (both in abstract and main content).

Mention the inclusion and exclusion criteria separately so it is made clear.

Provide the search terms / search string that’ll be used in in the initial search.

Please indicate weather a quality appraisal of the articles will be performed or not, with reasoning if not.

Avoid approximations within methodology and indicate what exactly would be done.

E.g.,

‘a search strategy from electronic search engines such as………….’

‘The review will focus on related topics such as HIV transmission, safe…..’

‘This review will not include certain types of publications such as’

Mention, in the analysis section, how the expected framework will be built using the data from the study.

Writing and typographic errors;

Following are examples only and please check for all such typos and revise.

- Introduction 1st paragraph line 11 – Full stop

- Introduction 4th paragraph – ‘The (38) has also proposed using psychosocial’

- World Health Organizations

Some places lack appropriate referencing;

E.g.,

Although specific components, such as the psychosocial approach recommended by the World Health Organizations (2023), have been proposed for intervention, the context and individual characteristics play significant roles in determining how to approach HIV prevention and care among adolescents (REF).

Some of the content is repetitive;

(e.g., last paragraph of the discussion). Check and revise the manuscript to correct such repetitive information.

Some content is redundant within sentences.

E.g., ‘To ensure up-to-date information, only studies published after 2013 will be included, and earlier ones will be excluded.’ Here, the ‘earlier ones will be excluded’ appear redundant.

Reviewer #2: The topic is very relevant and urgent in the fight against HIV.

The protocol is well written.

However, I had to read the protocol repeatedly trying to understand the methodology. While the reviewers propose to use the enhanced JBI methodology, the protocol lacks a review question, and related objectives. Rather, there a mention of an objective (p.g. 9 under outcome, and aim (p.g. 10 in the discussion), I suggest this should be included in the introduction.

7. PLOS authors have the option to publish the peer review history of their article (what does this mean?). If published, this will include your full peer review and any attached files.

Reviewer #1: No

Reviewer #2: **Yes: **Emmy Kageha Igonya

---

## [Author Response · Author response to Decision Letter 0]

12 Dec 2023

We have attached all responses of reviewers and editor in the uploaded letter as directed by the editor in document titled "Response to Reviewers."

---

## [Decision Letter · Decision Letter 1]

21 Feb 2024

PONE-D-23-24131R1Adolescent HIV Prevent and Care Framework: A Global Scoping Review Protocol-BSGH 006PLOS ONE

Dear Dr. Shamrock,

Thank you for submitting your manuscript to PLOS ONE. After careful consideration, we feel that it has merit but does not fully meet PLOS ONE’s publication criteria as it currently stands. Therefore, we invite you to submit a revised version of the manuscript that addresses the points raised during the review process.

We look forward to receiving your revised manuscript.

Kind regards,

Lara Vojnov

Academic Editor

PLOS ONE

Journal Requirements:

Reviewers' comments:

Reviewer's Responses to Questions

**Comments to the Author**

1. Does the manuscript provide a valid rationale for the proposed study, with clearly identified and justified research questions?

Reviewer #1: Yes

Reviewer #3: Partly

2. Is the protocol technically sound and planned in a manner that will lead to a meaningful outcome and allow testing the stated hypotheses?

Reviewer #1: Yes

Reviewer #3: Yes

3. Is the methodology feasible and described in sufficient detail to allow the work to be replicable?

Reviewer #1: Yes

Reviewer #3: Yes

4. Have the authors described where all data underlying the findings will be made available when the study is complete?

Reviewer #1: Yes

Reviewer #3: No

5. Is the manuscript presented in an intelligible fashion and written in standard English?

Reviewer #1: Yes

Reviewer #3: Yes

6. Review Comments to the Author

You may also provide optional suggestions and comments to authors that they might find helpful in planning their study.

Reviewer #1: The Authors have addressed the previous comments / suggestions mostly. However the minor comment 'Avoid approximations within methodology and indicate what exactly would be done' has not been properly addressed, despite being indicated as addressed in the response letter.

Reviewer #3: The rationale does not come out clearly. Authors need to succinctly explain why this undertaking is necessary.

Some of the statements need to be backed up with references. An example is stating that 'Among adolescents, HIV/AIDs remain the second leading cause of death globally'. Please check that this is correct, and the citation provided does not help much.

A few texts are difficult to interpret and, in some cases, come off as overstated. Some examples:

''Sex stigma in some communities deter adolescents from accessing HIV care - A bit difficult to comprehend.

Determining appropriate antiretroviral (ART) dosages for sustained suppression in adolescents is challenging - Does not come off as factual.

The World Health Organizations (2023) has also proposed using psychosocial techniques to support preventive measures for HIV among adolescents - any citation?''

Authors need to check the feasibility of outcomes. For instance, is it feasible to generate a comprehensive data analysis that would consider such diverse social factors such as geography, race, sex, age, gender, education,

religion, sexual orientation, occupation, nationality, marital status, socio-economic status, disability, and immigration status?

Finally, the authors speak to the dissemination of findings but not the ready availability of results and comprehensive data. Is there an existing repository that can house such data?

7. PLOS authors have the option to publish the peer review history of their article (what does this mean?). If published, this will include your full peer review and any attached files.

Reviewer #1: No

Reviewer #3: No

---

## [Author Response · Author response to Decision Letter 1]

5 Mar 2024

All reviewer comments have been addressed and file containing these responses have been attached to the submission.

---

## [Decision Letter · Decision Letter 2]

1 Jul 2024

Adolescent HIV Prevent and Care Framework: A Global Scoping Review Protocol-BSGH 006

PONE-D-23-24131R2

Dear Dr. Shamrock,

We’re pleased to inform you that your manuscript has been judged scientifically suitable for publication and will be formally accepted for publication once it meets all outstanding technical requirements.

Kind regards,

Graeme Hoddinott, Ph.D

Academic Editor

PLOS ONE

Additional Editor Comments (optional):

Reviewers' comments:

Reviewer's Responses to Questions

**Comments to the Author**

1. Does the manuscript provide a valid rationale for the proposed study, with clearly identified and justified research questions?

Reviewer #1: Yes

2. Is the protocol technically sound and planned in a manner that will lead to a meaningful outcome and allow testing the stated hypotheses?

Reviewer #1: Yes

3. Is the methodology feasible and described in sufficient detail to allow the work to be replicable?

Reviewer #1: Yes

4. Have the authors described where all data underlying the findings will be made available when the study is complete?

Reviewer #1: Yes

5. Is the manuscript presented in an intelligible fashion and written in standard English?

Reviewer #1: Yes

6. Review Comments to the Author

You may also provide optional suggestions and comments to authors that they might find helpful in planning their study.

Reviewer #1: Authors have addressed the previous edits / comments adequately.

Authors have addressed the previous edits / comments adequately.

7. PLOS authors have the option to publish the peer review history of their article (what does this mean?). If published, this will include your full peer review and any attached files.

Reviewer #1: No

---

## [Editor Report · Acceptance letter]

5 Jul 2024

PONE-D-23-24131R2 

PLOS ONE

Dear Dr. Shamrock, 

I'm pleased to inform you that your manuscript has been deemed suitable for publication in PLOS ONE. Congratulations! Your manuscript is now being handed over to our production team.

Kind regards, 

on behalf of

Dr. Graeme Hoddinott 

Academic Editor

PLOS ONE